# SYRAC: Synthesize, Rank, and Count

## Abstract

Crowd counting is a critical task in computer vision, with several important applications. However, existing counting methods rely on labor-intensive density map annotations, necessitating the manual localization of each individual pedestrian. While recent efforts have attempted to alleviate the annotation burden through weakly or semi-supervised learning, these approaches fall short of significantly reducing the workload. We propose a novel approach to eliminate the annotation burden by leveraging latent diffusion models to generate synthetic data. However, these models struggle to reliably understand object quantities, leading to noisy annotations when prompted to produce images with a specific quantity of objects. To address this, we use latent diffusion models to create two types of synthetic data: one by removing pedestrians from real images, which generates ranked image pairs with a weak but reliable object quantity signal, and the other by generating synthetic images with a predetermined number of objects, offering a strong but noisy counting signal. Our method utilizes the ranking image pairs for pre-training and then fits a linear layer to the noisy synthetic images using these crowd quantity features. We report state-of-the-art results for unsupervised crowd counting. As part of our commitment to fostering reproducibility within the field, we plan to release all synthetic datasets, code, and model checkpoints.

## 1 Introduction

Crowd counting is a crucial task in computer vision, finding application in downstream use cases like retail analysis, event management, and city planning. Conventionally, fully-supervised methods for crowd counting rely on density map annotations, where each person's location is marked with a point by an annotator. However, this annotation process is time-consuming. Cholakkal et al. (2020) estimated that, in simple scenes with low object density, it takes around 1.1 seconds to annotate each object. Whereas, Wang et al. (2020) found that annotating 2,133,375 pedestrians in complex and high density scenes across 5,109 images took 3,000 hours of human labor, which provides an estimate of 5.1 seconds per object. This represents a significant burden for the annotator.

To alleviate this issue, recent approaches have explored alternative strategies. Semi-supervised crowd counting involves fully annotating a subset, typically 5% to 10%, of the dataset. This reduces the annotation effort but still requires manual annotation for a significant portion of the data. Similarly, weakly supervised crowd counting strategies attempt to solve the problem with annotations that are faster to collect. For example, Yang et al. (2020) proposed learning global object count labels and D' Alessandro et al. (2023) proposed learning from inter-image ranking labels. Nevertheless, even with these strategies, some degree of annotation burden persists. Ideally, a crowd counting method should strive to eliminate the need for any manual annotation burden altogether.

Unsupervised crowd counting, which is typically defined as quantifying the number of people in an image without training on manually collected annotations (Liang et al., 2023a), is a challenging problem. Solutions must produce networks capable of identifying the number of objects in a scene across a wide range of conditions and object variations, without direct information about the quantity of objects present in the training data. To our knowledge, only two previous papers have approached this challenging problem. These approaches typically involve a pre-training step, where crowd quantity features are captured, and a count decoding phase, where the crowd count is extracted from the pre-trained features. CSS-CCNN (Babu Sam et al., 2022) approaches the problem by assuming that crowds naturally follow a power law distribution. They first train a feature encoder using a self-supervised proxy task and then utilize Sinkhorn matching to map the learned features to a prior

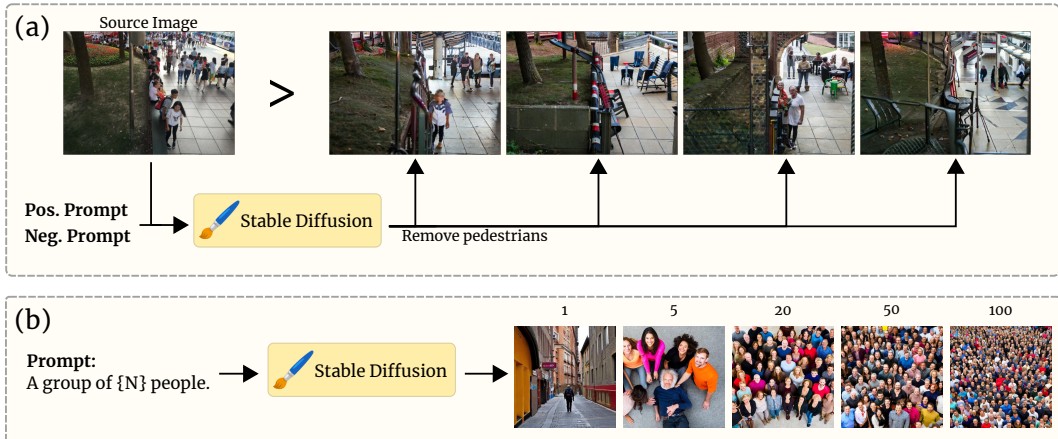

Figure 1: **Image Synthesis Overview.** (a) Pedestrians are removed from a real source image to produce a synthetic image and ranking label, which encodes a weak crowd quantity relationship. (b) Latent diffusion models can be used to generate synthetic images of crowds. However, specifying the number of pedestrians is often challenging and noisy. Listed above are the values of N.

distribution that adheres to a power law. CrowdCLIP (Liang et al., 2023a) extracts a crowd counting signal from a contrastive pre-trained vision-language model (CLIP). They employ a multi-modal ranking loss to learn crowd quantity features and then apply a progressive filtering strategy to select the correct crowd patches and match them to count intervals within text prompts. Both methods provide interesting approaches to the challenging unsupervised object counting problem.

One possible approach to eliminate the expenses associated with manual data annotation and accomplish unsupervised crowd counting is by employing generative models like Stable Diffusion (Rombach et al., 2022) to synthesize training data. Diffusion generative methods have shown promise in zero-shot recognition, few-shot recognition, and model pre-training (Shipard et al., 2023; He et al., 2023), but they have been mainly applied to object recognition rather than crowd counting. A straightforward approach might be to prompt the network to produce images with a specific crowd count, using explicit quantity labels, such as "*An image with 72 people*" with "72" used as the crowd count label. We refer to this as synthetic noisy count data. The process of generating this noisy synthetic crowd counting data is outlined in Figure 1-b. However, generating images with a precise count of objects using diffusion models is an unreliable process (Paiss et al., 2023; Cho et al., 2022). Figure 2 provides an example of the significant variance in the underlying count for synthesized crowd images. Therefore, alternative approaches are required to address the challenge of unsupervised crowd counting using synthetic data generated by such models.

We propose using diffusion-based image manipulation techniques (Meng et al., 2022; Rombach et al., 2022) to alter crowd counts in images by removing pedestrians, as demonstrated in Figure 1. Subsequently, we leverage these before-and-after images as a weak pairwise ranking signal for training a network with the ability to capture crowd quantity features. The primary challenge we address is the task of learning crowd counting from this synthetic ranking data and the synthetic noisy count data.

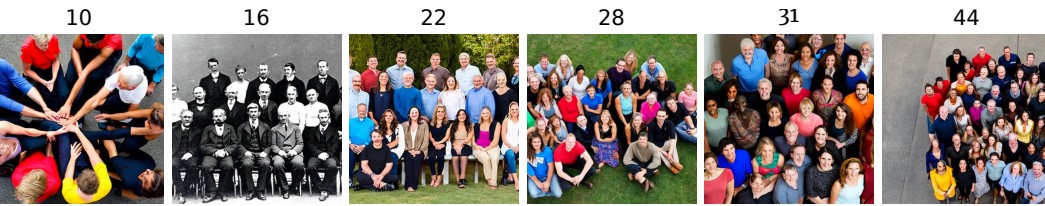

Figure 2: **Counting Noise.** All of the above images are synthesized by Stable Diffusion using the prompt "A group of 20 people." Listed above are the true counts, demonstrating a disparity between the expected count and true count.

We divide the learning process into two steps: pre-training and linear probing. In the pre-training phase, we train a Siamese network using the synthetic ranking dataset. The outcome of this phase is a network that encodes features related to crowd quantity. With this model we can better leverage the synthetic noisy count data. To accomplish this, we perform linear probing using the pre-trained features and the synthetic crowd images. As the pre-trained model captures crowd quantity information, we can constrain the set of possible functions to those specifically related to crowd quantity. The linear model then establishes a relationship between crowd quantity features and the noisy count, which reduces the risk of overfitting to label noise (Zheltonozhskii et al., 2022; Xue et al., 2022). In summary, we make the following contributions:

- A novel synthetic data generation strategy for unsupervised crowd counting
- A novel strategy for exploiting the synthetic ranking data and utilizing noisy synthetic counting data to extract true counts from the ranking network features
- We report SOTA performance on several crowd counting benchmark datasets for the unsupervised crowd counting task.

## 2 RELATED WORK

**Fully-Supervised Crowd Counting.** Crowd counting has seen significant success when learning from density maps, which are generated by convolving Gaussian kernels with dot map annotations (Zhang et al., 2016; Lempitsky & Zisserman, 2010; Li et al., 2018). GauNet (Cheng et al., 2022) employs locally connected Gaussian kernels to approximate density map generation and introduces a low-rank approximation approach with translation invariance. GLoss (Wan et al., 2021b) reframes density map estimation as an unbalanced optimal transport problem and introduces a generalized loss function tailored to learning from density maps. AMSNet (Wan et al., 2021a) adopts a neural architecture search-based approach and introduces a multi-scale loss to incorporate structural information effectively. ADSCNet (Bai et al., 2020) propose a strategy for density map self-correction by adaptively setting the dilation rate for convolutions. These strategies represent several interesting directions for directly utilizing density map annotations.

**Crowd Counting with Limited Annotations.** Semi-supervised object counting involves leveraging a small fraction (typically 5% to 10%) of fully-supervised density maps from the available dataset alongside a substantial volume of unlabeled examples (Zhao et al., 2020; Sam et al., 2019; Liu et al., 2020). Optimal Transport Minimization (Lin & Chan, 2023) minimizes the Sinkhorn distance to generate hard pseudo-labels from estimated density maps. Meng et al. (2021) employs a spatial uncertainty-aware teacher-student framework to learn from surrogate tasks on unlabeled data. Sindagi et al. (2020a) introduces an iterative self-training approach using a Gaussian Process to estimate pseudo ground truth on unlabeled data. Weakly-supervised object counting methods, on the other hand, rely on a weaker form of annotation that still necessitates manual collection but provides less information compared to fully-supervised examples. Yang et al. (2020) proposed a technique for learning exclusively from global object counts, incorporating a sorting loss that encourages the network to capture inter-example relationships across the dataset. In a different vein, D' Alessandro et al. (2023) employed annotated inter-image ranking pairs to enable learning from a diverse set of ranking images.

**Unsupervised Crowd Counting.** Eliminating the annotation burden from the crowd counting task is an incredibly challenging problem. There are two previous works which have approached this problem. CSS-CCNN (Babu Sam et al., 2022) introduces an unsupervised approach that incorporates self-supervised pre-training using image rotation as a training target. The authors make the critical assumption that crowd images adhere to a power law distribution and employ Sinkhorn matching to map the acquired features to this distribution. Importantly, the only form of supervision required in this approach is prior knowledge of the maximum crowd count and the parameters of the power law distribution. Liang et al. (2023a) propose CrowdCLIP, an unsupervised method which involves exploiting highly informative features present within CLIP, a pre-trained language-image model. They use CLIP as part of a pre-training step to learn from multi-modal ranking that uses ordered text prompts. They then introduce a filtering strategy for selecting image patches containing crowds and match features to text prompts representing crowd count categories. While no labeled annotations are applied within their training loss, the authors do utilize labeled data for performing

Source Image          Generated Images

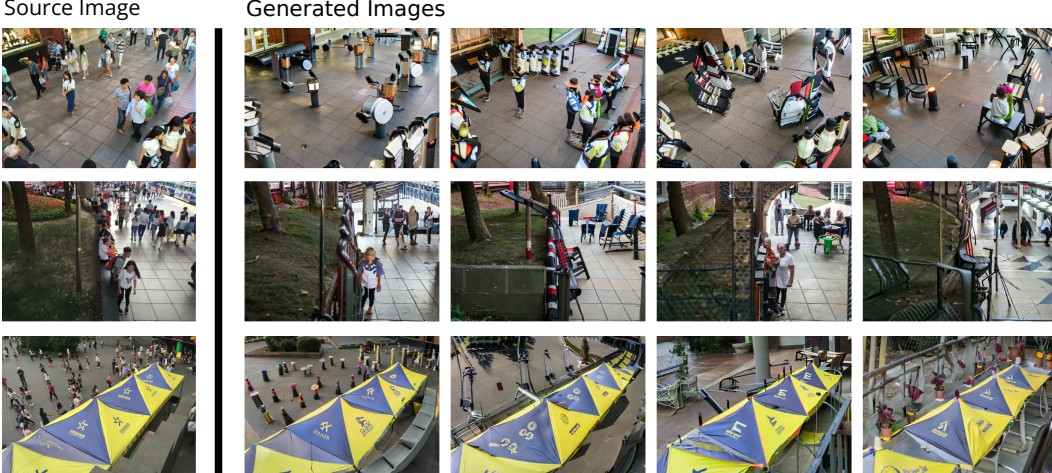

Figure 3: **Ranking Samples.** Synthetic images (right) generated from source images (left) from the ShanghaiTech B dataset retain details from the original image, but remove many or all pedestrians.

early stopping to select the optimal model[1]. It is important to note that early stopping is a regularization technique aimed at mitigating overfitting, and labeled data is typically unavailable for model selection in an unsupervised context. Consequently, we contend that while we draw comparisons between our method and CrowdCLIP, only our approach and CSS-CCNN remain strictly unsupervised, delineating a different annotation burden in the realm of unsupervised crowd counting.

**Generative Models.**   Stable Diffusion (SD) (Rombach et al., 2022) is a generative model designed for image synthesis, based on the concept of latent diffusion models. SD facilitates image generation through a multi-step process. To generate a synthetic image, a real image is encoded using a variational autoencoder, producing a compressed representation in a lower-dimensional latent space. Subsequently, Gaussian noise is iteratively applied to the latent features. To restore the meaningful features an iterative denoising mechanism, known as the reverse diffusion process, is employed. Finally, the new image is synthesized from the denoised features using a decoder. SD can be conditioned on text embeddings, typically obtained from a text-encoder like CLIP (Radford et al., 2021). This enables the model to generate images guided by user defined text prompts.

## 3   SYNTHETIC DATA GENERATION

Recent works have demonstrated that Stable Diffusion (SD) struggles with comprehending object quantity in text prompts (Paiss et al., 2023; Cho et al., 2022). In fact, SD generates an incorrect number of objects the majority of the time (Cho et al., 2022), and training text embedding models like CLIP to accurately count small quantities requires re-training on task specific data (Paiss et al., 2023). As an alternative approach, we propose utilizing an image-to-image latent diffusion model to remove pedestrians from real source images and thus obtain a object quantity ranking label. To achieve this, we employ positive and negative prompts to discourage the diffusion model from producing the objects of interest. This results in synthetic images with consistently fewer pedestrians compared to their source images, thanks to the noise injection process obscuring pedestrians and the prompts actively discouraging the re-introduction of those pedestrians in the synthetic image. We leverage this property of latent diffusion models to create a pairwise ranking signal, capturing a weak yet reliable crowd quantity relationship between the source image and the synthetic image.

Compared to directly synthesizing crowd count data from prompts, generating ranking data proves to be considerably more reliable. Although the ranking signal may be weaker, we show that it is a consistent and dependable crowd-quantity signal for pre-training counting models.

---

[1]Usage of early stopping is not documented in their paper (Liang et al., 2023a). However, by examining their code and via direct correspondence with the author, we verify their use of early stopping where they monitor performance using the ground truth count of the test set.

### 3.1 SYNTHETIC RANKING DATA

The goal of our synthetic data generation approach, is to create a ranking dataset wherein each training sample comprises a pair of images with a known count rank relationship (e.g. image A has a larger crowd count than image B). Our novel approach to generate such pairs utilizes SD to perform image-to-image synthesis to remove pedestrians from a real source image and generate a similar image with fewer pedestrians. Our approach has a similar motivation to intra-image ranking strategies for semi-supervised counting (Liu et al., 2018; Liang et al., 2023b), which involve selecting a sub-crop from a real source image, while exploiting that the sub-crop should contain an equal or smaller number of pedestrians compared to the original image. However, intra-image ranking tends to perform poorly when used as the only supervisory signal. Instead, our method benefits from retaining the scenes perspective and many of the scene features from the original source image. Further, multiple synthetic images can be generated per source image, which introduces small amounts of inter-image variance. This encourages the network to rely on only the crowd quantity features to rank image pairs. In the subsequent sections, we will delve into further detail, providing a comprehensive explanation of these steps.

**Synthesis.** We are interested in generating synthetic images by removing pedestrians from real source images to produce a pairwise ranking dataset. Suppose we are provided with a dataset containing images of crowds without any annotations:

$$\mathcal{D}_{\mathtt{crowd}}^{\mathtt{real}} = \{x_i^{\mathtt{real}}\}^{N_{src}},$$

where $x_i^{\mathtt{real}}$ represents an image containing an unknown number of pedestrians, $c_i^{\mathtt{real}}$, and there are $N_{src}$ total examples in the dataset. We would like to apply a latent diffusion model, $G$, to our images with the goal of generating similar images but with fewer pedestrians. We apply $G$ as follows:

$$x_i^{\mathtt{syn}} = G(x_i^{\mathtt{real}}, t_{\mathtt{prompt}}, t_{\mathtt{neg}}, \epsilon),$$

where $x_i^{\mathtt{syn}}$ is a synthetic image generated using $x_i^{\mathtt{real}}$, $t_{\mathtt{prompt}}$ is a text prompt which guides the generation process, $t_{\mathtt{neg}}$ is an additional negative prompt to guide the synthetic image away from specific outcomes, and $\epsilon$ represents the stochastic nature of the latent diffusion model. The image $x_i^{\mathtt{syn}}$ contains an unknown number of objects $c_i^{\mathtt{syn}}$, but we would like to select prompts which lead to the following training examples:

$$\mathcal{D}_{\mathtt{rank}}^{\mathtt{syn}} = \{(x_i^{\mathtt{real}}, x_i^{\mathtt{syn}}), c_i^{\mathtt{real}} \geq c_i^{\mathtt{syn}}\}^{N_{syn}}.$$

This synthetic dataset possesses several intuitively useful properties. Firstly, it provides image pairs which share a weak object quantity relationship. Secondly, the synthetic images retain many of the image features present in the real source images, helping to suppress spurious solutions. Thirdly, the synthetic images often still contain some relevant objects, introducing a challenging ranking problem where ordering the images requires understanding the object quantity. Lastly, relevant objects are replaced with irrelevant objects, creating competing features that encourage the network to focus on the most relevant information.

For each real source image, $x_i^{\mathtt{real}}$, we generate four unique synthetic images, which leads to a synthetic dataset with size $N_{syn} = 4 * N_{src}$. Figure 3 depicts examples for three source images.

**Prompt Selection.** We manually select the prompts used for generating the synthetic ranking dataset. This is a common strategy for zero-shot classification methods and unsupervised crowd counting methods that rely on CLIP (Wortsman et al., 2022; Liang et al., 2023a). While there may be a more principled approach for selecting prompts for generating counting data, we followed a very simple approach. We manually tested a few prompts for a single image and selected the one which looked best. We set $t_{\mathtt{prompt}}$ to "An empty outside space. Nobody around. High resolution DSLR photography." and we set $t_{\mathtt{neg}}$ to "people, crowds, pedestrians, humans, 3d, cartoon, anime, painting". The negative prompt is critical in this setup, as it encourages the latent diffusion model to generate an image that has as few relevant objects as possible while also appearing realistic. Our prompt selection was further guided by reviewing some common practices considered useful by SD users. However, the different prompts we reviewed did not produce significantly different images.

### 3.2 NOISY SYNTHETIC COUNTING DATA

The goal of our noisy synthetic data generation approach is to produce synthetic images which have approximate but noisy count labels. However, noisy synthetic counting data presents a substantial

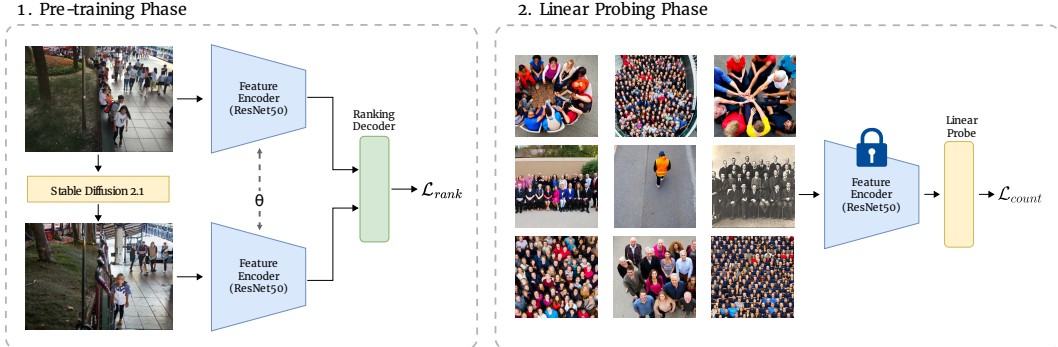

Figure 4: **Methodology**. Our two step process involves pre-training an image encoder using synthetic ranking pairs, and then linear probing the model using noisy synthetic counting data.

challenge due to the notable disparities between the expected count implied by the prompt and the actual count within the generated image. This inherent discrepancy complicates the learning process, as neural networks can easily succumb to overfitting in the presence of label noise. However, we demonstrate that there is inherent merit in incorporating these noisy examples alongside the synthetic ranking data, allowing us to perform a form of linear probing on the pre-trained features.

We utilize SD to perform text-to-image synthesis, using text to guide the model towards generating synthetic images with a specific number of pedestrians. We deploy SD as follows:

$$\{x_i^{\mathtt{syn}}, c_i^{\mathtt{prompt}}\} = G(t_{\mathtt{prompt}},\ t_{\mathtt{neg}},\ \epsilon),$$

where $c_i^{\mathtt{prompt}}$ is the number of pedestrians specified in the prompt, whereas the true count, $c_i^{\mathtt{real}}$, is unknown to us but correlated with $c_i^{\mathtt{prompt}}$. This generates a dataset of noisy synthetic examples:

$$\mathcal{D}_{\mathtt{count}}^{\mathtt{syn}} = \{x_i^{\mathtt{syn}}, c_i^{\mathtt{prompt}}\}^{N_{noisy}}.$$

We set $t_{\mathtt{prompt}}$ as "A group of $\{N\}$ people. High angle", where $N = \{1, 5, 10, 50, 100, 200\}$. We set the negative prompt $t_{\mathtt{neg}}$ as "B&W, cartoon, anime, 3D, watercolor, artistic, geometric." We generate 60 images for each value of $N$, producing 360 examples which we use for linear probing. Additionally, we include 60 images using the prompt "an empty urban environment", with $t_{\mathtt{neg}}$ as "Humans, people, pedestrians, crowds, B&W, cartoon, anime, 3D, watercolor, artistic, geometric". We consider these images to have 0 pedestrians.

## 4 Unsupervised Crowd Counting

We split the unsupervised crowd counting problem into two steps. The first step, which is highlighted in Figure 4-1, involves pre-training for a pairwise ranking network using a ranking loss. This pre-training phase aims to harness the synthetic ranking dataset to extract features closely correlated with pedestrian quantity. Subsequently, in the second step, also shown in Figure 4-2, we focus on training a linear model using the noisy counting data, ultimately yielding a joint model capable of crowd quantification in images.

### 4.1 Synthetic Ranking Pre-Training

In Section 3 we outlined strategies for generating image ranking pairs that have a crowd quantity based ordering that we use for model pre-training. We seek to exploit these labels to learn features that correlate with the crowd count. To this end, we employ a Siamese model, outlined in Figure 4-1.

The core architecture comprises two primary components: a feature encoding model denoted as $f_\theta : x \to z \in \mathbb{R}_+^{2048}$ and a ranking decoder head represented by $h_\Theta : z \to c \in \mathbb{R}_+$. Here, $z$ represents the encoded features of an image $x$, and $c$ is a proxy count used for the ranking task. For each pair of images $x_i^{\mathtt{real}}$ and $x_i^{\mathtt{syn}}$, we pass them through the joint encoder-decoder to compute proxy counts $\hat{c}_i^{\mathtt{real}}$ and $\hat{c}_i^{\mathtt{syn}}$.

Table 1: **Crowd Counting Performance.** Results are reported for the test set of each respective dataset. We compete directly with unsupervised methods (None) but we highlight fully-supervised methods (DMap) for comparison. Missing results are due authors not evaluating on all datasets.

| Method | Venue | Labels | SHB MAE | SHB MSE | JHU MAE | JHU MSE | SHA MAE | SHA MSE | QNRF MAE | QNRF MSE |
|---|---|---|---|---|---|---|---|---|---|---|
| ADSCNet | CVPR'20 | DMap | 6.4 | 11.3 | - | - | 55.4 | 97.7 | 71.3 | 132.5 |
| AMSNet | ECCV'20 | DMap | 6.7 | 10.2 | - | - | 56.7 | 93.4 | 101.8 | 163.2 |
| GLoss | CVPR'21 | DMap | 7.3 | 11.7 | 59.9 | 259.5 | 61.3 | 95.4 | 84.3 | 147.5 |
| GauNet | CVPR'22 | DMap | 6.2 | 9.9 | 58.2 | 245.1 | 54.8 | 89.1 | 81.6 | 153.7 |
| CrowdCLIP | CVPR'23 | None† | 69.3 | 85.8 | 213.7 | 576.1 | 146.1 | 236.3 | 283.3 | 488.7 |
| CSS-CCNN | ECCV'22 | None | - | - | 217.6 | 651.3 | 197.3 | 295.9 | 437.0 | 722.3 |
| Ours (2x2) | | None | **49.0** | **60.3** | **194.0** | **583.9** | **196.0** | **295.2** | **390.0** | **697.5** |

† CrowdCLIP utilizes ground truth fully supervised labels to perform early stopping. As such, we do not consider this to be a strictly unsupervised approach.

To compute the ranking loss, we follow an approach similar to previous methods (Liu et al., 2018; Liang et al., 2023a; D' Alessandro et al., 2023). Our ranking loss is formulated as follows:

$$\mathcal{L}_{rank} = \mathbb{E}_{(x_i^{\text{real}}, x_i^{\text{syn}}) \in \mathcal{D}_{rank}^{syn}} \left[ -\log(\sigma(\hat{c}_i^{\text{real}} - \hat{c}_i^{\text{syn}})) \right],$$

where $\sigma$ represents the sigmoid function. Beyond this, we also apply a non-negativity constraint to the parameters of the ranking decoder head. The features given to the ranking decoder are passed through a ReLU activation function, and are thus non-negative. Given this, the non-negativity constraint prevents the ranking decoder from reducing proxy counts by making irrelevant features negative. Our aim is to discourage the network from fixating on irrelevant attributes that could inadvertently aid in identifying images with a lower proxy count, and thus encourage the model to prioritize meaningful attributes during crowd counting tasks.

## 4.2 SYNTHETIC NOISY LINEAR PROBING

In section 3 we outlined a strategy for generating a synthetic noisy counting dataset. These standalone synthetic samples are, by themselves, unsuitable for direct utilization in training a counting model due to the potential for overfitting to label noise. Furthermore, they are entirely synthetic in nature, resulting in a discernible domain gap. However, our methodology leverages the pre-trained image encoder. This encoder remains frozen during the training process, and we subsequently fine-tune a linear layer, $g_\rho : z \to c$, on top of it. The pre-training process plays a pivotal role in mitigating the inherent noise within the synthetic counting examples, which is a phenomena observed in previous works on object recognition (Zheltonozhskii et al., 2022; Xue et al., 2022). This is achieved by constraining the feature selection process to focus exclusively on those features that encapsulate pertinent crowd quantity information. The loss function for this step is given as:

$$\mathcal{L}_{count} = \mathbb{E}_{(x_i^{\text{syn}}, c_i^{\text{prompt}}) \in \mathcal{D}_{count}^{syn}} \left[ (\hat{c}_i^{\text{syn}} - c_i^{\text{prompt}})^2 \right].$$

We then use the pre-trained network and the linear layer jointly to predict the crowd count in images.

## 4.3 INFERENCE

In prior unsupervised counting methodologies (Liang et al., 2023a; Babu Sam et al., 2022), a common approach involved dividing images into a grid of patches. This patch-based division served to simplify the counting problem and manage the number of objects per image effectively. Typically, the choice of the number of patches in the grid was made empirically, with a tendency to opt for more patches when dealing with datasets featuring denser crowds. In our method, we introduce a maximum count constraint for the synthetic noisy counting dataset, limiting it to 200 objects. This constraint arises from the increase in label noise as crowd quantity increases. Given that crowd counting datasets often encompass counts ranging from hundreds to thousands, we adopt this strategy to maintain relevance to real-world scenarios. Furthermore, we delve into a thorough exploration of the impact of patch size in Table 2.

Table 2: **Patch Size.** A patch size of 2 provides SOTA result, although it is not always optimal.

| Patch Size | SHB | | JHU | | SHA | | QNRF | |
|---|---|---|---|---|---|---|---|---|
| | MAE | MSE | MAE | MSE | MAE | MSE | MAE | MSE |
| 1 | **38.9** | **55.8** | 215.0 | 654.3 | 259.5 | 411.5 | 392.6 | 692.7 |
| 2 | 49.0 | 60.3 | **194.0** | **583.9** | **196.0** | **295.2** | 390.0 | 697.5 |
| 3 | 85.8 | 94.4 | 226.42 | 584.64 | 239.6 | 329.7 | 355.5 | 641.5 |
| 4 | 160.1 | 166.8 | 267.8 | 610.1 | 336.4 | 453.2 | **351.9** | **611.3** |

Table 3: **Pre-Training Strategy.** Comparing synthetic ranking to other pre-training strategies, as well as end-to-end training on the noisy synthetic counting dataset without pre-training.

| Pre-training | SHB | | JHU | | SHA | | QNRF | |
|---|---|---|---|---|---|---|---|---|
| | MAE | MSE | MAE | MSE | MAE | MSE | MAE | MSE |
| No Pre-training | 113.6 | 145.8 | 304.4 | 778.7 | 372.6 | 498.2 | 671.0 | 997.9 |
| ImageNet | 54.4 | 89.9 | 287.5 | 1077.5 | 243.11 | 370.0 | 535.6 | 932.6 |
| Intra-image Rank | 59.9 | 89.2 | 209.9 | 619.9 | 226.0 | 365.0 | 486.1 | 841.8 |
| Ours (2x2) | **49.0** | **60.3** | **194.0** | **583.9** | **196.0** | **295.2** | 390.0 | 697.5 |

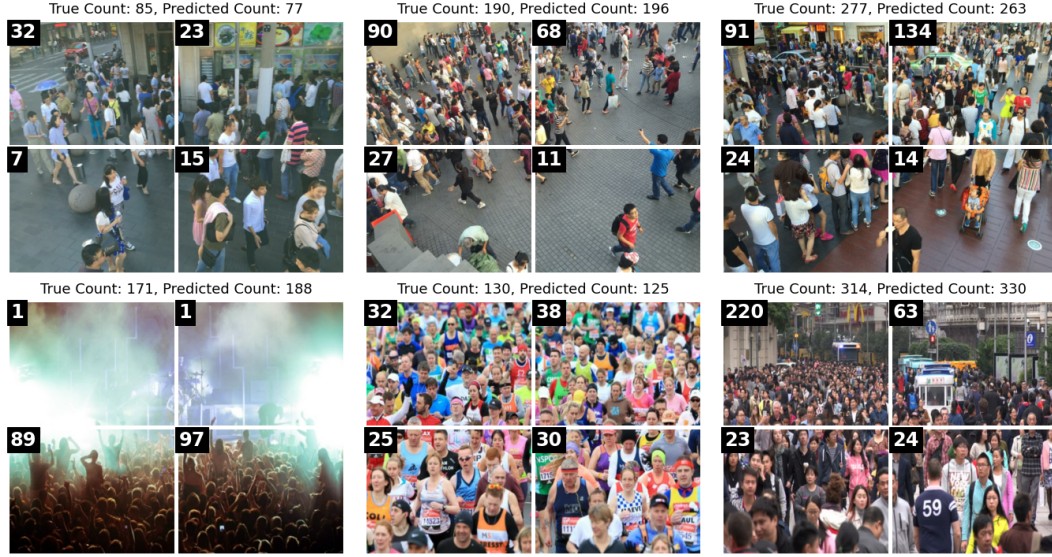

Figure 5: **Qualitative Results**. Our method performs exceptionally well across a range of crowd sizes. Top: patch-wise predictions on ShanghaiTechB. Bottom: patch-wise predictions on JHU++. Each image patch is annotated with its predicted count.

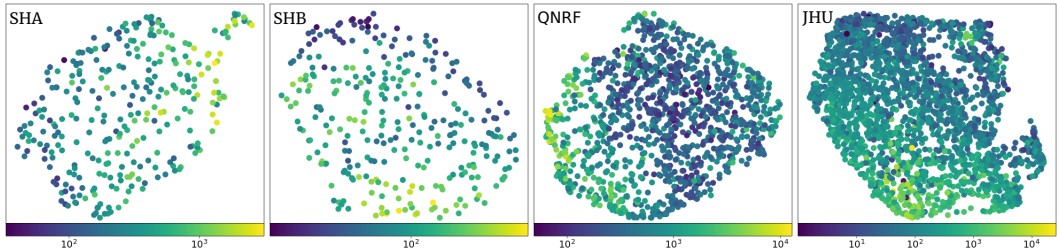

Figure 6: **Pre-Training features**. A 2-dimensional UMAP embedding for ShanghaiTechA, ShanghaiTechB, JHU-Crowd++, and UCF-QNRF features demonstrates an underlying crowd-count based ordering. Points are colored by their respective count.

## 5 Experiments & Results

**Implementation.** In our implementation, we employ a Siamese network as the ranking encoder, with ResNet50 (He et al., 2016) serving as the underlying architecture. We select a model pre-trained on ImageNet to harness its feature extraction capabilities. The rank decoder is designed as a single fully connected layer with a non-negativity constraint. We train our model for 40 epochs, utilizing the Adam optimizer with a learning rate set to $5e^{-5}$. To maintain consistency in our dataset, we resize all images to a uniform size of (640, 853, 3). During training, we incorporate straight-forward geometric and visual augmentations, including random flipping and adjustments to image brightness. For the data generation process, we rely on Stable Diffusion 2.1. When performing image-to-image generation, we set the strength parameter to 0.45. Throughout all image generation procedures, we maintain a fixed guidance scale of 7.5 and carry out optimization for 50 steps. We set aside 15% of the synthesized ranking data as a validation set for performing model selection.

**Datasets.** We benchmark on 4 datasets: ShanghaiTechA (SHA) (Zhang et al., 2016), Shang-haiTechB (SHB) (Zhang et al., 2016), UCF-QNRF (Idrees et al., 2018), and JHU-Crowd++ (Sindagi et al., 2020b) which contain, respectively, 482, 716, 1535, and 4372 images with min/avg/max per-image crowd counts of 33/501/3139, 9/123/578, 49/815/12865, and 0/346/25791.

**Unsupervised Crowd Counting.** Table 1 illustrates the superior performance of our method in comparison to the previous state-of-the-art (SOTA) unsupervised crowd counting methods. Our approach achieves a mean absolute error of 49.0 on the SHB dataset, 194.0 on the JHU dataset, 196.0 on the SHA dataset, and 390.0 on the QNRF dataset. Our method significantly outperforms the prior SOTA for unsupervised crowd counting and even beats CrowdCLIP on SHB and JHU, despite their use of ground truth counting labels for early stopping. This is likely due to SHB and JHU having lower average counts, where our method excels.

**Impact of Patch Size.** We investigate the influence of patch size on our counting results. We analyze different partitioning schemes, including 1x1, 2x2, 3x3, and 4x4 grids. By referencing Table 2 and Table 1, we can see that a patch size of 2 consistently yields SOTA performance across all datasets. However, better results are achieved with fewer patches on datasets with lower average counts and with more patches on datasets with higher average counts.

**Impact of Pre-Training.** We investigate the impact of different feature learning strategies for our pre-training step. We investigate ImageNet features and investigate intra-image ranking, similar to previous semi-supervised crowd counting strategies (Liu et al., 2018). Table 3 highlights that our method provides superior performance over other pre-training strategies. Further, it demonstrates that end-to-end training with noisy synthetic counting data, without any strategy to mitigate the label noise, leads to disastrous results. Figure 6 demonstrates that our method results in a network that learns an underlying ordering that corresponds to the true ordering of the crowd counting datasets.

## 6 Conclusions

In this study, we have tackled the challenging task of unsupervised crowd counting in computer vision. Our solution leverages latent diffusion models to generate synthetic data, thus eliminating the annotation burden. However, these models face challenges in reliably understanding object quantities, which can result in noisy annotations when tasked with generating images with specific object quantities. To mitigate this issue, we employed latent diffusion models to create two types of synthetic data: one by removing pedestrians from real images, which yields ordered image pairs with a weak but reliable object quantity label, and the other by generating synthetic images with a predetermined number of objects, offering a strong but noisy counting label.

Further, we deploy a two-step strategy for extracting crowd counts from these datasets. In doing so, we demonstrate our method's superiority over the SOTA in unsupervised crowd counting across multiple benchmark datasets. Our work not only significantly alleviates the annotation burden asso-ciated with crowd counting but outlines a new direction for unsupervised crowd counting and what can be achieved in this field.

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
