# SUPPLEMENTARY MATERIAL

## A  DOMAIN SHIFT

In this section, we present experiments offering evidence that our synthetic ranking dataset closely resembles the true distribution compared to GCC (Wang et al., 2019). To assess the domain shift, we pass 500 randomly sampled images from each dataset through a ResNet50 network pre-trained on ImageNet for natural image classification. These resulting features help us analyze each distribution.

Initially, we generate 2D UMAP, t-SNE, and PCA embeddings for each dataset's points and visualize them in Figure 1. From this visualization, it's evident that the synthetic ranking dataset exhibits significant overlap with the real crowd counting dataset. Conversely, the GCC dataset shows limited overlap with the real crowd counting dataset.

Moving further, we employ two metrics to quantify distances between empirical feature distributions. Firstly, we compute feature means for each distribution and measure the L2 distance between these means for each dataset. We report an L2 distance of 56.2 between the feature means of the synthetic ranking dataset and the real crowd counting dataset, and an L2 distance of 110.3 between the feature means of the GCC and the real crowd counting dataset.

Secondly, we reduced the original (N, 2048)-dimensional datasets to (N, 3) dimensions using PCA and then calculate the Chamfer distance between them. Here, we report a distance of 36.4 between the synthetic ranking dataset and the real crowd counting dataset, and a distance of 57.8 between the GCC and the real crowd counting dataset.

These analyses support the assertion that our synthetic ranking dataset aligns more closely with the true distribution compared to GCC.

## B  SYNTHETIC DATASET QUALITY

### B.1  SYNTHETIC RANKING DATASET

To substantiate the reliability of the synthetic ranking data, we examine 50 ranking pairs from each dataset. This inspection revealed minimal discrepancies: ShanghaiTechB produced no incorrectly ranked examples, while ShanghaiTechA exhibited only one. QNRF and JHU++ presented slightly higher instances of two incorrectly ranked examples each. Moreover, to improve our assessment, we employed a fully-supervised DM-COUNT (Wang et al., 2020) model to evaluate crowd count estimations across 500 real and synthetic ranking pairs from each dataset. The resultant accuracy rates

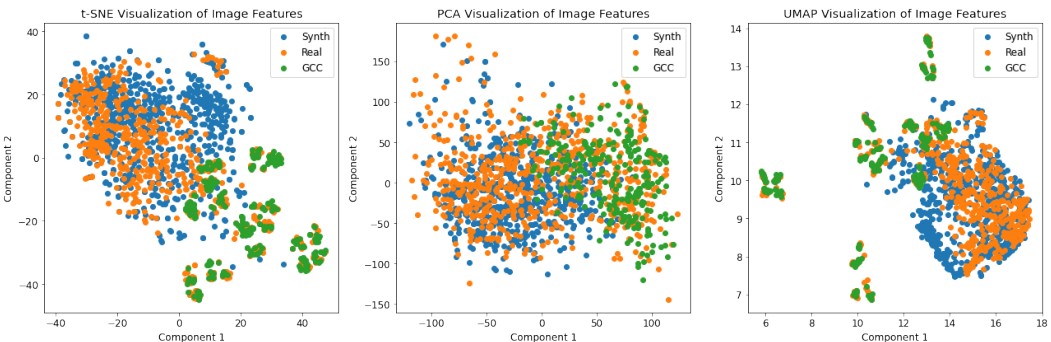

Figure 1: **Domain Gap**. A 2-dimensional visualization of the features from the GCC dataset, our synthetic ranking dataset, and the real crowd counting dataset with respect to a ResNet50 network pre-trained on ImageNet. We use t-SNE, PCA, and UMAP to demonstrate qualitatively that GCC has a larger domain gap relative to the real crowd counting dataset than our synthetic ranking data.

Table 1: **Estimated Statistics.** We use a pre-trained counting model (DM-Count), trained on the NWPU dataset, to estimate the count in each noisy synthetic counting image. We report the predicted mean, standard deviation, maximum count, and minimum count across all synthetic images.

| Prompt Count | Mean | StD | Min | Max |
| --- | --- | --- | --- | --- |
| 1 | 0.53 | 1.00 | 0 | 5 |
| 5 | 1.90 | 1.38 | 0 | 5 |
| 20 | 18.81 | 11.33 | 2 | 78 |
| 50 | 71.85 | 59.25 | 10 | 353 |
| 100 | 150.07 | 118.50 | 19 | 763 |
| 200 | 245.80 | 93.55 | 37 | 484 |

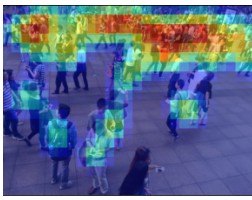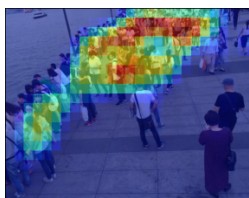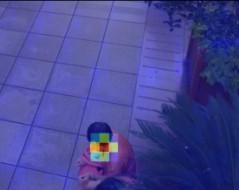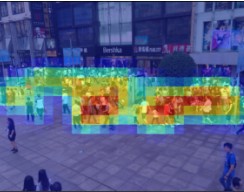

Figure 2: **Localization**. Heatmap depicting the network's activation in regions with crowds, derived from features of the last convolutional layer using our method on ShanghaiTechB dataset samples.

were high: 99.6% for ShanghaiTechB, 99.2% for ShanghaiTechA, 97.2% for QNRF, and 90.2% for JHU++. These findings affirm the credibility of the synthetic ranking data generation process.

### B.2 NOISY SYNTHETIC COUNTING DATASET

Table 1 showcases results obtained using the fully-supervised DM-Count model (Wang et al., 2020), trained on the NWPU dataset, to estimate counts in our noisy synthetic counting images. It provides predicted mean, standard deviation, maximum, and minimum counts across these images.

We observe that the predicted mean closely aligns with the count label in the prompt, suggesting that our generation process is working reasonably well. However, as crowd counts increase, the standard deviation widens significantly. This divergence between predicted and prompt counts intensifies with larger crowd sizes, indicating heightened label noise. This implies that while the model's mean predictions initially match the provided count labels, it struggles more with matching the prompt as crowd sizes grow.

## C QUALITATIVE EVALUATION OF LOCALIZATION

Even though our method doesn't rely on location-based annotations like density maps, it effectively localizes crowds. To showcase this qualitatively, we employ our pre-trained network to compute features for various samples from the ShanghaiTechB dataset. Subsequently, we derive the channel-wise average of the features obtained from the final convolutional layer, visualizing this as a heatmap (refer to Figure 2). These examples demonstrate the network's pronounced activation in areas populated by crowds. This observation strongly suggests its efficacy in crowd localization, all achieved without accessing any location-based annotations.

## D EXTENSION TO NOVEL OBJECT CATEGORIES

Our methodology extends beyond crowd counting to encompass a diverse range of novel object categories. Our primary objective is to showcase the adaptability and efficacy of our approach in handling entirely new object classes with minimal adjustments required. To illustrate this, we apply our methodology to penguins, vehicles, dogs, and tomatoes.

Real Source Image   Synthetic Images with Penguins Removed

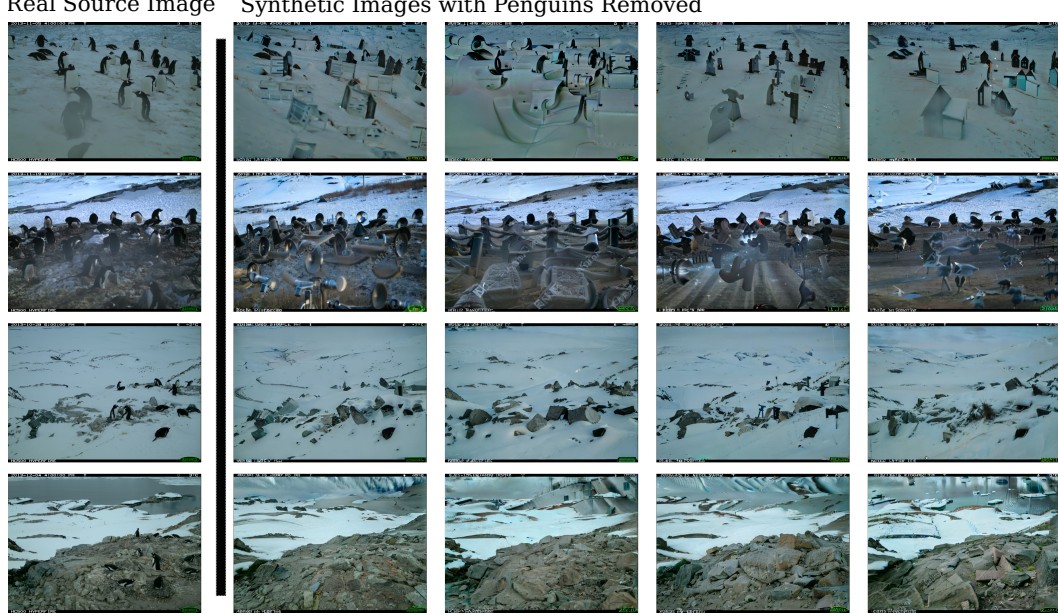

Figure 3: **Penguins Synthetic Removal**. Comparison between real image of penguins (left) and synthetic images (right) generated by our method, demonstrating penguin removal while preserving scene features.

Real Source Image   Synthetic Images with Vehicles Removed

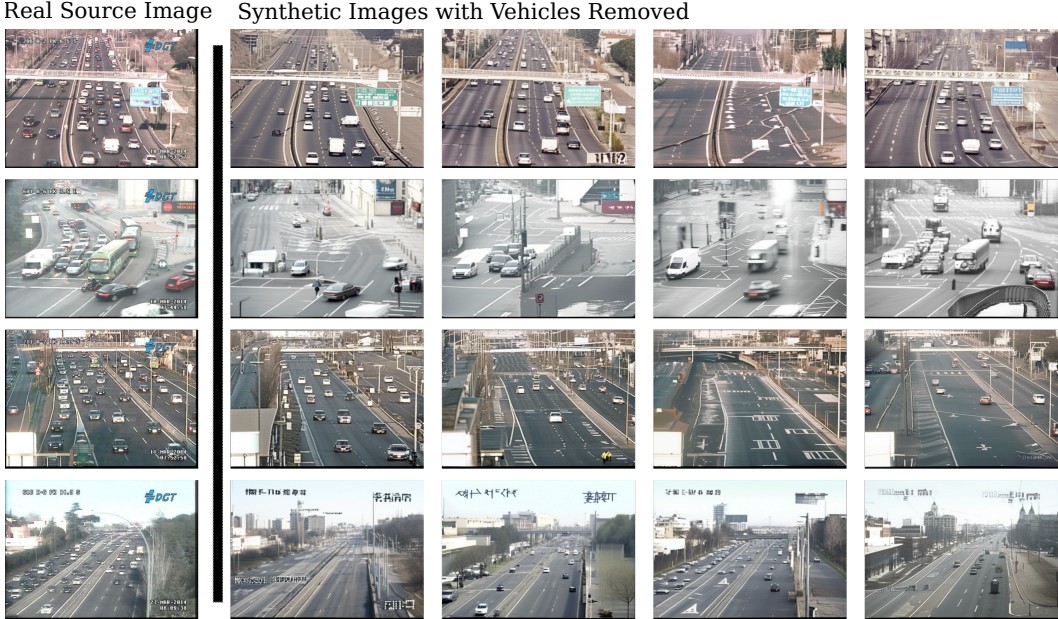

Figure 4: **Vehicles Synthetic Removal**. Comparison between real image of vehicles (left) and synthetic images (right) generated by our method, demonstrating vehicle removal while preserving scene features.

### D.1 QUALITATIVE ANALYSIS OF SYNTHETIC RANKING EXAMPLES

The Penguins dataset Arteta et al. (2016) and TRANCOS dataset Guerrero-Gómez-Olmedo et al. (2015) serve as commonly used object counting benchmarks outside the realm of crowd counting. Leveraging these datasets, we demonstrate that our strategy for synthetic object removal across these categories. Figures 3 and 4 present qualitative results that showcase our method's ability

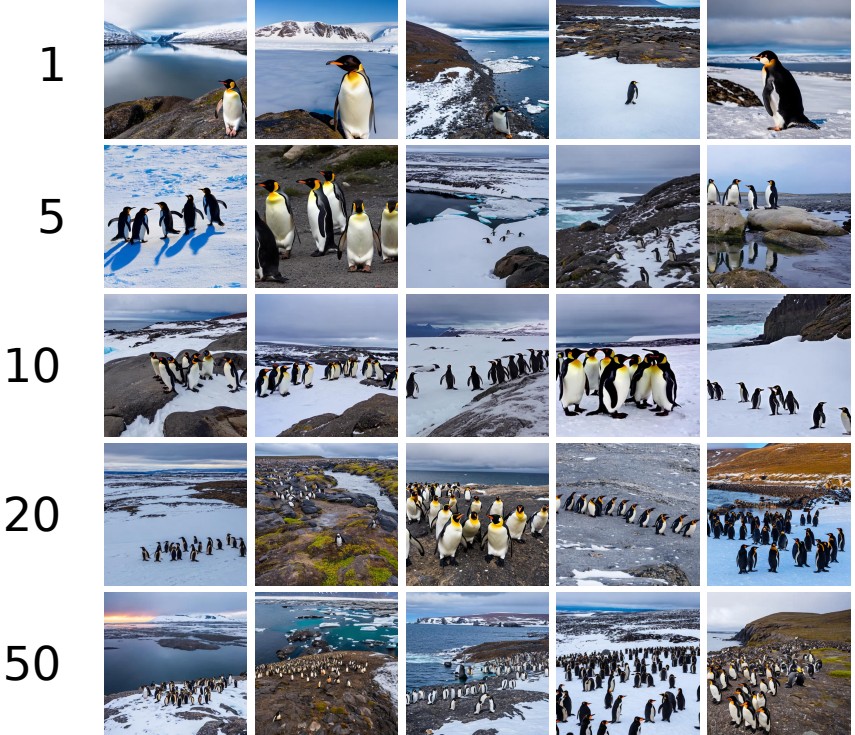

Figure 5: **Penguins Noisy Synthetic**. Synthetic images of penguins for prompt labels: {1, 5, 10, 20, 50}

to accurately remove objects while preserving essential scene features present within the original image.

### D.2 QUALITATIVE ANALYSIS OF NOISY SYNTHETIC COUNTING EXAMPLES

Expanding our approach to generate noisy synthetic counting examples for novel categories—dogs, penguins, vehicles, and tomatoes—we generate 15 examples per category using the same methodology employed for crowd counting. From these, we select the top 5 examples in each category to highlight our method's capacity to produce images reflecting the specified object counts. Given that TRANCOS dataset has an average of 36 vehicles, and Penguins has an average of 7 penguins, we restrict our synthesis to prompts with counts set to N = {1, 5, 10, 20, 50}. Visualizations of penguins (Figure 5), vehicles (Figure 8), tomatoes (Figure 7), and dogs (Figure 6) underscore our ability to generate noisy synthetic counting examples across various object categories. While these examples exhibit label noise similar to previous noisy synthetic crowd counting instances, they maintain approximate correctness, showcasing the versatility of our approach across diverse object categories.

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

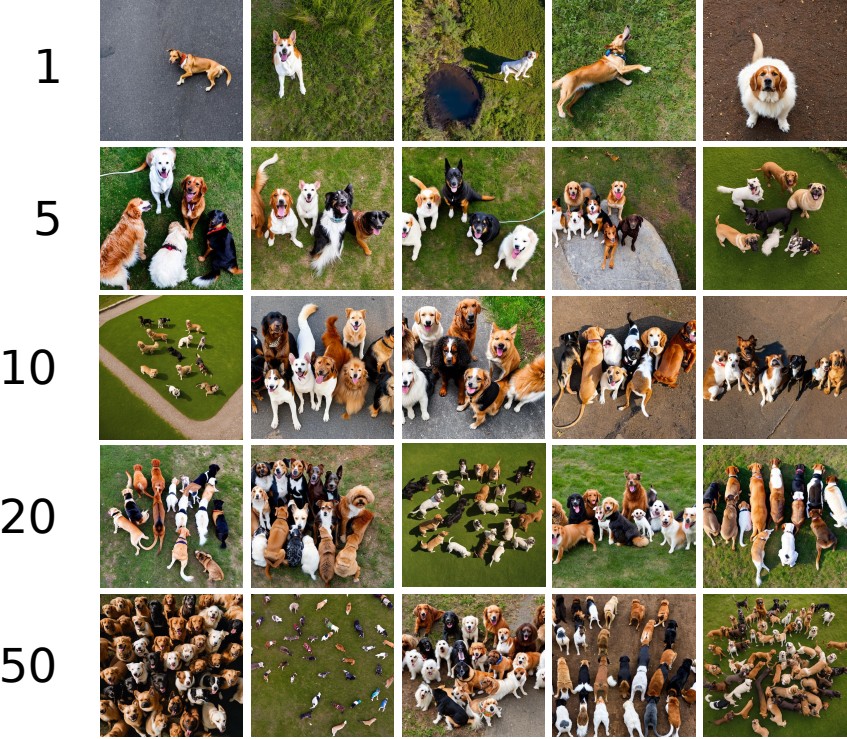

Figure 6: **Dogs Noisy Synthetic**. Synthetic images of dogs for prompt labels: {1, 5, 10, 20, 50}

Qi Wang, Junyu Gao, Wei Lin, and Yuan Yuan. Learning from synthetic data for crowd counting in the wild. In *Proceedings of IEEE Conference on Computer Vision and Pattern Recognition (CVPR)*, pp. 8198–8207, 2019.

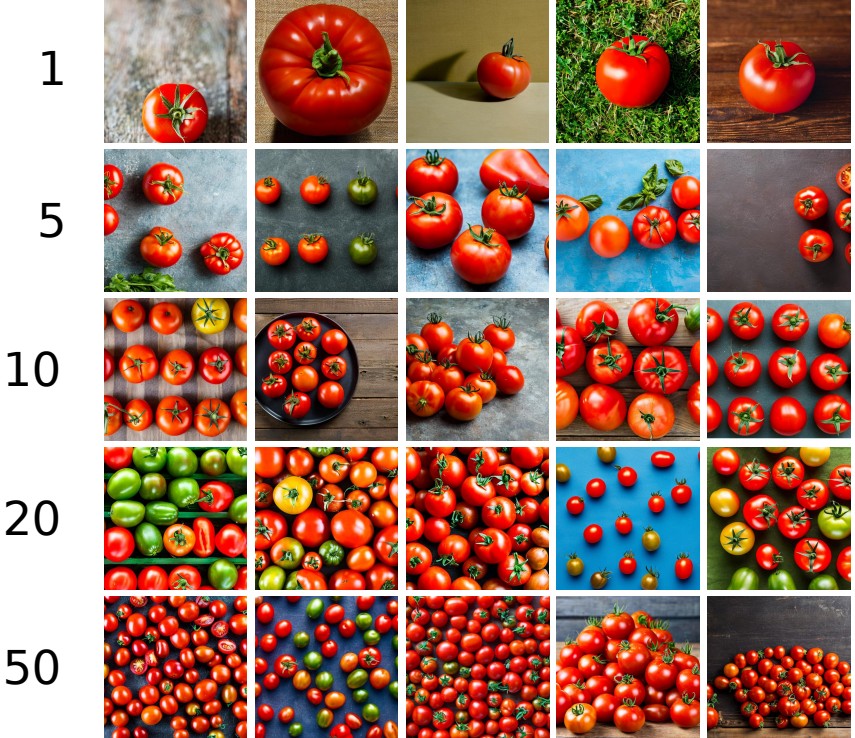

Figure 7: **Tomato Noisy Synthetic**. Synthetic images of tomatoes for prompt labels: {1, 5, 10, 20, 50}

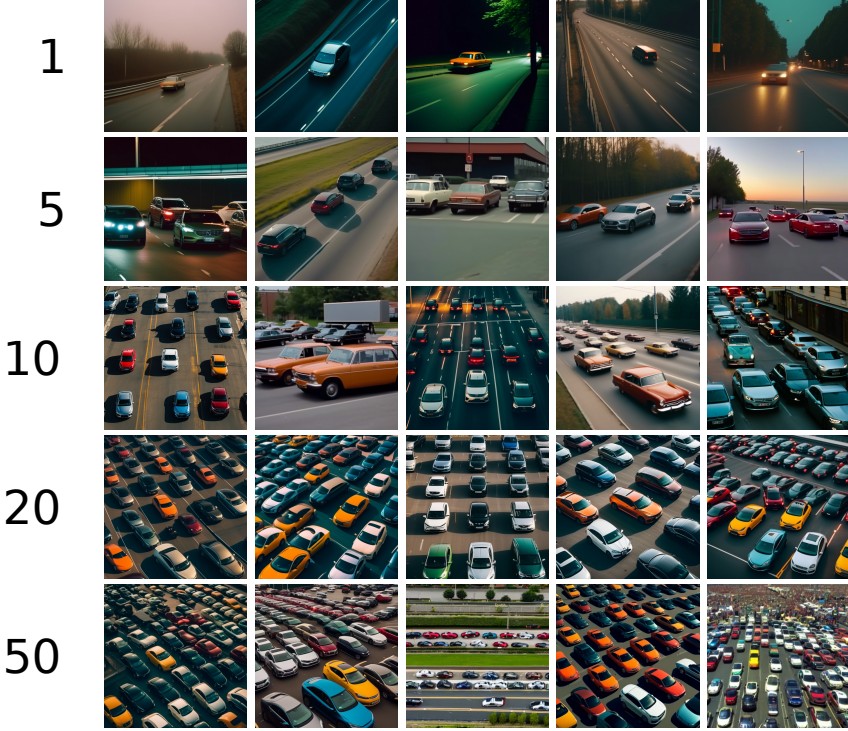

Figure 8: **Vehicles Noisy Synthetic**. Synthetic images of vehicles for prompt labels: {1, 5, 10, 20, 50}