# OpenReview forum: "SYRAC: Synthesize, Rank, and Count"
_ICLR.cc/2024/Conference — Submitted to ICLR 2024_

### Official Review · Reviewer_QJht · 2023-10-29

**Soundness:** 2 fair
**Presentation:** 2 fair
**Contribution:** 2 fair
**Rating:** 5
**Confidence:** 5

**Summary:**

This paper focuses on the unsupervised crowd counting task, a critical yet challenging task. To achieve this goal, the authors use latent diffusion models to create two types of synthetic data and then utilize the ranking image pairs for pre-training and fit a linear layer to the noisy synthetic images using these crowd quantity features. Experiments conducted on five datasets demonstrate the effectiveness of the proposed method.

**Strengths:**

(a) Using stable diffusion to generate the crowd dataset is a good idea, providing a new perspective for this area.
(b) This paper is written well and easy to follow

**Weaknesses:**

1. For the fully supervised part, the authors only discuss the density-based crowd counting methods. In other words, many localization-based methods should be discussed, making the related work more comprehensive.

2. The authors have pointed out that the prompt count is not reliable but using it as the GT count directly during the training phase. It makes me confused. I think it would be better to rank the generated 60 images using the pre-trained backbone first. Secondly, fine-tune the GT count according to the ranking results. Specifically, image A and image B are generated using the same prompt count 20. However, ranking results present that image A contains fewer persons than image B, so the GT count of image A could be fine-tuned to be smaller than the GT count of image B.

3. I understand that the input of the generation process is complete images without cropping, but the inference process uses image patches as input. There may be resolution gaps. How about cropping the original images into patches in the generation process instead?

4. There is a lack of quantitative analysis about the reliability of the generation process. Specifically, the authors can sample n source images for generation and statistics on the percentage of images where the objects were successfully removed.

5. The authors think the ranking information is reliable, and the prompt count is relatively unreliable. Thus, the authors pre-train the backbone using the ranking information and freeze the backbone during the training phase to resist the prompt count noise. I agree that the ranking information is more reliable. However, I am not sure it is necessary to fix the backbone as only fine-tuning the linear layer may limit the learning potential on the prompt count, which is considered ground truth. There could be an ablation study on fine-tuning the backbone during the training phase.

6. Since the current method is still significantly lower than CrowdCLIP, the authors think the early stop used in CrowdCLIP might be unfair. So I would like to know the performance under early stop.

7. The motivation to synthesize the ranking crowd image is still unclear since one can utilize the existing datasets to generate the ranking image pairs, like CrowdCLIP.

**Questions:**

see weakness

---

> ### Author Response · Authors · 2023-11-17
> **Rebuttal for Reviewer QJht**
>
> We thank the reviewer for their valued insights. We would like to address some of their concerns:
> 1. **Counting vs. Detection:** We appreciate this insight. We would be happy to expand our discussion to emphasize how crowd counting is a different task than detection and thus can rely on different features, for two reasons: 1) Counting is a more relaxed task than detection in that the former does not require the localization of objects like the latter, 2) counting may leverage features that respond to multiple quantities of objects (not only to one object) in different spatial constellations, e.g. clusters of pedestrians, whereas in object detection, the features delineate a single object. Additionally, typical object detection datasets will have around 1 to 20 objects in an image. Crowd counting datasets tend to have an average of more than 100 pedestrians per image, and density based methods tend to outperform detection based methods in these scenarios.
> 2. **Ranking Noisy Synthetic Counts:** The goal of our synthetic ranking pre-training strategy is to mitigate label noise by narrowing down the feature set to those specifically linked to crowd quantity before initiating fine-tuning with the noisy synthetic counting data. While we could pseudo-label the noisy synthetic data using the ranking network, the ranking network may itself be unreliable due to a domain shift between the synthetic ranking dataset and the noisy synthetic counting dataset.
> 3. **Cropping before Synthesis:** We did run experiments where we cropped a portion of a source image and generated a synthetic ranking pair that way, similar to your idea. However, this performed poorly in practice. We suspect that was due to the way that pedestrians are distributed in crowd counting images, resulting in many crops that had no crowds in them. As an example, the perspective of an image might cause the top half of the image to contain only sky or buildings or other unrelated features while the bottom half contains the crowd. This setup would result in comparing a real image with no crowds to a synthetic image with no crowds, this potentially impacts feature learning.
> 4. **Reliability of Ranking Labels:** We agree that it would be good to provide a more concrete analysis of the reliability of the synthetic ranking data. We manually inspected 50 ranking pairs from each dataset to estimate the accuracy. For ShanghaiTechB, there were no incorrectly ranked examples. For ShanghaiTechA there was 1 incorrectly ranked example. For QNRF there were 2 incorrectly ranked examples. And, for JHU++, there were 2 incorrectly ranked examples. Additionally, we use a pre-trained crowd counting model (DM-COUNT [1]) to estimate the crowd count for 500 real and synthetic ranking pairs to provide an additional estimate of the reliability of the synthetic ranking data. The estimated accuracy is 99.6% for ShanghaiTechB, 99.2% for ShanghaiTechA, 97.2% for QNRF, and 90.2% for JHU++. These additional experiments corroborate the idea that the synthetic ranking data is highly reliable.
> 5. **Fine-tuning Backbone:** We agree that it would be useful to have an ablation study examining different strategies for tuning the pre-trained network with the noisy counting data. We did run experiments where we included the noisy synthetic counting data during pre-training with the synthetic ranking data. However, that did not achieve good results.
> 6. **Comparison to CrowdCLIP:** Our method outperforms CrowdCLIP on 2 of the 4 datasets. We had considered using early stopping in a similar fashion to CrowdCLIP. However, during our discussion with the authors of CrowdCLIP (which we have documented for reference), we had found that they used the **test** set for early stopping. We could not justify this as an experiment, as it would seriously contaminate the results.
> 7. **Motivation for Synthetic Ranking:** We do utilize existing datasets to generate the ranking pairs, as those images are the source of the synthetic ranking data. Methods like CrowdCLIP generate ranking pairs by performing “intra-image ranking”. In Table 3 we demonstrate that intra-image ranking underperforms compared to our synthetic ranking strategy.
>
> [1]. Wang, Boyu, et al. "Distribution matching for crowd counting." Advances in neural information processing systems 33 (2020): 1595-1607.

---

> ### Comment · Reviewer_QJht · 2023-11-22
>
> 1. Are the results chosen from the latest epoch?
>
> 2. See the log from CrowdCLIP, and we can find that CrowdCLIP might not utilize early stops since all models are trained on 100 epochs.

---

> > ### Author Response · Authors · 2023-11-22
> > **regarding CrowdCLIP**
> >
> > We thank the reviewer for their follow up question. We would like to reiterate that we have verified through correspondence with the first author of CrowdCLIP that they have performed early stopping using the test set to achieve their results. The authors did not report this in their paper. We have shared this correspondence with the program chairs so that it can be shared privately without de-anonymizing ourselves as authors.
> >
> > However, it is also clear that this is how they arrive at their results when inspecting the materials they shared publicly for reproducibility. To provide some additional evidence which is publicly available, please refer to [this link](https://onedrive.live.com/?authkey=%21AC%2D%2DlrsQWrW5JJI&id=255D1679C866D64F%21879&cid=255D1679C866D64F) where the authors release a checkpoint for a model trained on the shanghaitech A dataset, where the name of the checkpoint is “epoch=43-val_mae_metric=144.8791.ckpt”. Then please refer to [this link](https://onedrive.live.com/?authkey=%21AC%2D%2DlrsQWrW5JJI&id=255D1679C866D64F%21886&cid=255D1679C866D64F), where the authors release their training logs in a file titled “metrics.csv”. We can confirm from the checkpoint name and the log file that the author is selecting a model at epoch 43 with the best validation performance.

---

> ### Comment · Reviewer_QJht · 2023-11-23
>
> Thanks for the response.
> I would like to know the following information:
>
> (1) Are the results chosen from the latest epoch in the SYRAC?
>
> (2) I have checked the log of CrowdCLIP. However, if we also choose a fixed epoch (e.g., 50) and then utilize the final epoch as the final result, their results are still higher than SYRAC. For example, the 50th epoch's result is 150 MAE of CrowdCLIP on Shanghai PartA, significantly better than the proposed method. This is the reason why I ask the problem (1).
>
> (3) The training epoch is a hyperparameter. I think that the authors utilize 40 epochs for training, which might also based on experiential tuning.

---

> > ### Author Response · Authors · 2023-11-23
> > **Selecting a fixed epoch for CrowdCLIP**
> >
> > We thank the reviewer for their follow up.
> > 1. As we state in section 5 of the paper: "We set aside 15% of the synthesized ranking data as a validation set for performing model selection." We perform model selection using a hold out set of synthetic ranking data (rather than ground truth counting data from the test set).
> > 2. The authors have released additional checkpoints for their model trained on QNRF, SHA, and SHB which are available in their OneDrive folder linked above. They do not release checkpoints for JHU. They have chosen epoch 14 for QNRF, epoch 43 for SHA, and epoch 6 for SHB. I provide a table below showing a few difference scenarios for comparing results across different epochs. Our method beats 2 of the 3 results in all circumstances. In fact, there is no fixed epoch that they report where they outperform us on more than one dataset. And given that our method generally outperforms their method on the JHU dataset (as reported in their paper), it's clear that SYRAC outperforms CrowdCLIP on 3/4 datasets for every fixed epoch.
> > 3. The training epoch is certainly a hyperparmeter. However, tuning it using the test set introduces supervision. Further, it's not clear that there is a fixed epoch that works for each dataset. I would also like to refer the reviewer to this CVPR 2020 paper[1] which overviews the problem of using a fully-supervised validation set for hyperparameter tuning, which has serious implications for weakly supervised object localization methods.
> >
> > | Method    | SHB (mae) | SHA (mae) | QNRF (mae) |
> > | -------- | ------- | ------- | ------- |
> > | SYRAC (ours) | 49.0    | 196.0    | 390.0    |
> > | CrowdClip (epoch 50) |  93.8    | 149.4    | 554.7    |
> > | CrowdClip (epoch 100)    | 124.3    | 223.7    | 542.1   |
> > | CrowdClip (epoch 6)    | 68.4  | 213.4    | 420.1     |
> > | CrowdClip (epoch 14)    | 72.8    | 243.0    | 283.8    |
> > | CrowdClip (epoch 43)    | 95.9    | 144.9    | 425.0    |
> >
> > [1] Choe, Junsuk, et al. "Evaluating weakly supervised object localization methods right." Proceedings of the IEEE/CVF Conference on Computer Vision and Pattern Recognition. 2020.

---

### Official Review · Reviewer_rbvA · 2023-10-31

**Soundness:** 3 good
**Presentation:** 3 good
**Contribution:** 2 fair
**Rating:** 3
**Confidence:** 5

**Summary:**

This paper introduces an unsupervised counting method that utilizes latent diffusion models to create synthetic data. The approach involves two unsupervised techniques: first, removing pedestrians from actual images, resulting in ranked image pairs that provide a ranking loss of object quantity. Second, generating synthetic images with a predetermined number of objects, which gives a noisy but related counting label.

**Strengths:**

- The idea of utilizing a stable model to generate synthetic images seems feasible.
- The paper introduces two strategies: a weak but reliable object quantity signal and a strong but noisy counting signal. This approach seems quite reasonable, as it can potentially complement and enhance the model's performance.

**Weaknesses:**

- What is the rationale behind the setting of N, which is the crowd count to generate synthetic images? What is the quality of the generated images? Is it possible to provide a measure of variance to assess the feasibility of this method?
- There are only six categories for N. Why not train the model by a classification task? In situations where the labels are not stable, the classification task seems to be able to maintain a relatively high level of accuracy.
- The synthetic images do not include images with 0 crowd count. Does this method have the capability to handle datasets that consist of a large portion of (background) images with no people, such as NWPU?
- How does the computational cost of generating synthetic images using the diffusion model compare to that of other unsupervised counting models?
- There are some repetitions in the references.

**Questions:**

-  Figure 6 illustrates that the features exhibit an underlying crowd-count-based ordering. However, it would be more convincing if features from supervised counting models could be provided for comparison.
- In Table 3, the methods proposed in the paper actually include ImageNet pretraining. What is the performance when combining ImageNet pretraining with intra-image ranking?
- Does the ranking loss merely train the model to distinguish between real and synthetic images?

---

> ### Author Response · Authors · 2023-11-17
> **Rebuttal for Reviewer rbvA**
>
> We are grateful for your comprehensive feedback on our paper. We would like to address the points that you have raised.
> 1. **Quality & Settings for Noisy Synth Images:** To assess the quality + variance of our synthetic crowd counting image dataset, we've included a table below. The table reports the quality by using a fully-supervised model to estimate, for each synthetic image, the count and compares it to the desired count dictated by the prompt. The table reveals noise across count predictions for each count category. Despite this noise, the average predicted count closely aligns with the given count in the prompt.
> With regards to the count category selections in N, we used the following rationales for setting N =  {1, 5, 10, 20, 50, 100, 200}
>     - **Exponential:** We select values from small to large in an approximately exponential fashion. We chose 200 as a maximum because we use patching to split the image into regions with smaller counts (<200) during inference.
>     - **Computational Cost:** We avoid generating images from a more granular count range (i.e. 1, 2, 3, 4, …, 200) due to the increased computational cost this would incur.
>    - **Simplicity:** Our aim wasn't to optimize for specific prompts. While there might exist combinations of N that yield better performance, determining these without introducing validation data, thereby adding supervision, is challenging. We see this as a strength of our method, as the selection of N is based on simple heuristics, and yet it still works.
> 2. **Regression vs. Classification:** The count categories chosen for N = {1, 5, 10, 20, 50, 100, 200} represent a subset of counts among positive unbounded integers. Given the nature of our task, predicting counts intuitively aligns with regression rather than classification. We did consider different methods for fitting the pre-trained features to the noisy crowd counting data, including classification. We found that nothing performed better than linear regression. However, it’s possible that we did not explore this sufficiently and there may be a formulation where classification outperforms regression.
> 3. **Zero Count Images:** The synthetic images do include images with 0 crowd count. We didn’t explicitly show this in Figure 1. However, we do explain our strategy for generating images with 0 crowd count in the last two sentences of section 3.2.
> 4. **Computational Cost:** The question of computational cost is an important one, as generating synthetic data does incur an additional computational cost beyond the burden of training the network. We generated all synthetic data on a single GPU, and we also trained our models on a single GPU.
> 5. **Repititions:** Thank you, noted.
> 6. **Comparison to Full-Supervised Features:** This is a great point, thank you! We included Figure 6 to corroborate the idea that the pre-training features capture a quantity based distribution. However, it would be useful to further explore that distribution and compare it to features learned by different methods with different supervision budgets. We expect that more supervision would lead to similar but better organization of the latent space with regards to object count.
> 7. **ImageNet Pre-training:** In fact, we already leverage ImageNet pre-trained features for intra-image ranking. In Table 3, the performance for intra-image ranking is achieved by starting from ImageNet pre-trained features.
> 8. **Real vs. Synth Spurious Correlations:** This is a critical question. Deep learning performance normally suffers due to spurious correlations, and so we expect that our method may be prone to these spurious correlations, as well. However, we rely on the following evidence to suggest that our method produces a network which learns something coherent beyond the separation of real from synthetic:
>     - Our method outperforms other unsupervised methods, and our pre-training strategy outperforms other pre-training strategies.
>     - Figure 6 demonstrates a quantity based distribution.
>     - There is significant domain overlap between the synthetic and real distributions, as the synthetic images are generated using real images as a source. We provide a measure of this domain overlap in the supp. materials.
>     - In sec 5, we mention the use of a non-negativity constraint in the final layer of the network (+ ReLU activation), which prevents the weights and output from becoming negative. Thus, we expect that it will be harder for the network to exploit spurious correlations, as it cannot set the output of the network to be negative when synthetic features are present.
>
> That said, we could explore ways to further ensure that we mitigate learning features that distinguish real from synthetic images by producing ranked pairs of images where both images in each pair are synthetic (e.g. remove objects then remove some more; or by using SD to synthesize images then removing objects from those synthetic image), or by using domain adversarial learning.

---

> > ### Author Response · Authors · 2023-11-17
> > **Table for Noisy Synthetic Crowd Counting statistics**
> >
> > The table below is produced by using DM-Count [1], which is trained on the NWPU dataset (with full supervision using GT counts), to estimate the count in each noisy synthetic counting image. We report the predicted mean, standard deviation, maximum count, and minimum count across all synthetic images. We can see that for images with higher prompt counts, the predicted count can differ significantly from the prompt count. This suggests that label noise accumulates in larger crowd counts. The table is also reproduced in the supplementary material.
> >
> > | Prompt Count | Mean   | StD    | Min | Max |
> > |--------------|--------|--------|-----|-----|
> > | 1            | 0.53   | 1.00   | 0   | 6   |
> > | 5            | 1.90   | 1.38   | 0   | 5   |
> > | 20           | 18.81  | 11.33  | 2   | 78  |
> > | 50           | 71.85  | 59.25  | 10  | 353 |
> > | 100          | 150.07 | 118.50 | 19  | 763 |
> > | 200          | 245.80 | 93.55  | 37  | 484 |
> >
> > [1]. Wang, Boyu, et al. "Distribution matching for crowd counting." Advances in neural information processing systems 33 (2020): 1595-1607.

---

### Official Review · Reviewer_98cL · 2023-11-01

**Soundness:** 3 good
**Presentation:** 3 good
**Contribution:** 3 good
**Rating:** 6
**Confidence:** 5

**Summary:**

This paper presents an unsupervised crowd counting approach based on synthetic data. Specifically, it generates synthetic data through stable diffusion with a selected prompt and then employs the rank loss and a count loss for prediction. The excellent experimental results demonstrate the advantages of the proposed unsupervised method.

**Strengths:**

The idea is novel and the experimental results demonstrate the advantages of the proposed unsupervised method.

**Weaknesses:**

A deep analysis of the experimental results is not provided.

**Questions:**

1. Could the author provide a more comprehensive explanation for Figure 6? Both small and large counts are distributed throughout the entire space in the QNRF dataset. It is difficult to interpret the UMAP results without any explanation.
2. What is the advantage of generating synthetic data via stable diffusion, especially when compared with the large synthetic dataset GCC[1]? Both approaches are label-free, but GCC contains more detailed count and localization information. Additionally, [1] achieves better counting performance than the proposed method when there are no human-labeled annotations. It would be helpful to clarify the specific advantages of this paper.
3. Although this is an unsupervised method, it would be valuable to understand whether the pre-training phase performs as expected. The authors could randomly select pairs of images from SHA/SHB/QNRF to determine accuracy or probability and analyze cases in which it failed. Furthermore, the accuracy should be compared with a similar method presented in Liu et al[2].
4. The impact of patch size is only presented in the table. Could the authors provide a deeper analysis and discussion on the reasons for the observation that different patch sizes lead to different performance?"

[1] Wang, Qi, et al. "Learning from synthetic data for crowd counting in the wild," CVPR, 2019.
[2] Liu, Xialei, et al. "Leveraging unlabeled data for crowd counting by learning to rank," CVPR, 2018

---

> ### Author Response · Authors · 2023-11-17
> **Rebuttal for Reviewer 98cL**
>
> We thank the reviewer for their useful feedback and helpful questions. We would like to address the points that the reviewer has raised.
> 1. **Explanation for Figure 6.:** Figure 6 illustrates the UMAP representation of the training images based on the pre-trained synthetic ranking features, with points color-coded according to the logarithm of the ground truth count. While interpreting dimensionality reduction results can be complex, we included this figure to demonstrate that there is a coherent quantity-based structure to the latent space learned during the pre-training step. We can interpret this qualitatively by noticing clustering of similarly colored points, which suggests that images with similar counts tend to have similar features.
> 2. **GCC vs. SYRAC:** There are several advantages to using synthetic data generated by Stable Diffusion even when GCC already exists.
>     - **Labor Burden:** producing 3D synthetic counting data is not labor free, despite not requiring manual annotations. It requires acquiring new 3D assets for each new object category whereas our method can extend to any novel category that is recognized by Stable Diffusion. And, it requires designing complex 3D scenes as well as environmental conditions such as rain or limited light. Our approach does not carry this burden.
>      - **Domain Shift:** GCC exhibits a significant domain shift when compared to real crowd counting datasets, given that it is produced from 3D assets. Our method maintains a smaller domain shift by using real images as a seed for the synthetic ranking images and also due to the fact that Stable Diffusion is trained using natural images. We have added details to the supplementary material where we provide a measure of domain overlap between GCC and the synthetic ranking datasets, which demonstrates that our synthetic ranking data is closer to the real distribution than the GCC dataset.
>      - **Different Requirements:** While we currently underperform when compared to GCC based methods, we are also working on a fundamentally different problem. GCC requires human labor to produce, whereas our method does not. GCC provides density map annotations, whereas ours does not. Finally, methods that utilize GCC typically focus on domain adaptation due to the large domain gap between GCC and real datasets, whereas our method produces data that is already close to the true distribution.
> Given these reasons, we believe our paper serves as a valuable contribution to the research community
> 3. **Comparison with Liu et al.:** We do include intra-image ranking as a pre-training strategy in Table 3, which is the self-supervised component of their work. It is worth clarifying that Liu et al. do not perform self-supervised crowd counting, as their method relies on fully-supervised crowd counting data as the primary training signal. In row 3 of Table 3, we only include the results from the self-supervised component of Liu et al.’s method, which demonstrates that our method consistently outperforms intra-image ranking as a pre-training signal for each dataset.
> 4. **Patch Size:** To understand the impact of patch size, we consider two points:
>      - **Label Noise:** There is significant noise in the true underlying count for synthetic crowd counting data produced by Stable Diffusion. This noise increases as the expected count embedded within the prompt increases (see table below). Thus, we expect that prediction noise will be concentrated around images with larger underlying counts. To counter this, we adopt patching during inference, aiming to reduce the object count within each patch. Our findings (Table 2) indicate that increased patching benefits higher count datasets, improving performance by mitigating noise. However, this strategy may compromise images with smaller crowd counts, as larger objects get fragmented across patches. This was apparent in a performance drop among those images.
>      - **Limited Representation for Dense Crowds:** The synthetic ranking generation process often removes many objects in dense crowds, resulting in limited ranking examples for densely crowded images. Consequently, our model might struggle to learn features specific to highly dense crowds. Splitting these images into smaller patches during inference may align them with a distribution where the model performs more effectively.

---

> > ### Comment · Reviewer_98cL · 2023-11-19
> > **GCC vs. SYRAC**
> >
> > 1. SYRAC can extend to any novel category, but this paper does not show it. This paper is only for crowd counting.
> > 2. GCC contains domain shift, but it can provide detailed localization annotation. The domain shift is addressed well in many publications following GCC. SYRAC is close to the real domain. However, it cannot provide localization information. I also noticed that SYRAC cannot provide images containing dense crowds, i.e, thousands of humans, which is the major scene in crowd counting.
> > 3. I did not see any human labor that GCC requires. Could the author give me some details about human labor to avoid that I ignore some details?
> > 4. About the performance. The author should notice that the counting model trained on GCC performs better than yours even if the domain adaptation method is not applied:
> > |pre-train dataset| SHA (MAE/MSE)| SHB(MAE/MSE) | QNRF(MAE/MSE)|
> > |:-:|:-:|:-:|:-:|
> > | SYRAC | 196.0 / 295.2 | 49.0 / 60.3 | 390.0 / 697.5 |
> > | GCC (No-Adpt) | 160.0 / 216.5 | 22.8 / 30.6 | 275.5 / 458.5 |
> > | GCC (GAN-Adpt) | 123.4 / 193.4 | 19.9 / 28.3 | 230.4 / 384.5 |
> > experimental results come from [1].
> >  -----
> >
> > Based on the above, I think SYRAC has the following parts that should be improved:
> > 1. Extending to any novel category. If you say this is an advantage, you should show it in the paper;
> > 2. SYRAC can provide localization information and generate images containing thousands of humans freely;
> > 3. The performance is better than GCC without adaptation.
> >
> > I think the response will be recognized if any of the above points is achieved.
> >
> > ----
> > [1] Wang, Qi, et al. "Learning from synthetic data for crowd counting in the wild," CVPR, 2019.

---

> > > ### Author Response · Authors · 2023-11-21
> > > **GCC vs. SYRAC**
> > >
> > > We are grateful to the reviewer for their additional feedback. We would like to address the reviewers' concerns.
> > > 1. **Novel Categories**: We have added two sets of qualitative results to the supplementary materials. First, we add synthetic ranking examples for the penguins dataset[1] and the trancos dataset[2]. Then we add noisy synthetic counting examples for tomatoes, dogs, penguins, and vehicles. These examples provide evidence that our method can extend to novel categories. Previous unsupervised methods have only explored crowd counting, thus we do not have unsupervised methods to compare to. Given this, we are currently running experiments to compare with previous fully-supervised methods. We will update the supplementary materials with these results once available.
> > > 2. **GCC Comparison**: We would like to elaborate on the comparisons to GCC with respect to localization and dense crowd synthesis:
> > >     - **Localization of Crowds**: While GCC provides density map annotations, which result in a network that can provide location information, this is not strictly a necessary output for crowd counting. The ultimate goal is to acquire the global count. Density maps are used because they lead to improved performance. Unsupervised crowd counting methods, such as CrowdCLIP, do not localized crowds. Further, we have provided a figure in the supplementary material which demonstrates qualitatively the capacity for our network to localize crowds without being trained on any location based annotations. For the ShanghaiTechB dataset, we visualize the channel-wise mean of the feature maps for the final convolutional layer of our network. While these are not density maps, in that they do not explicitly integrate to the count, they do clearly demonstrate that our model localizes crowds and our model appears to somewhat capture the density of the crowds.
> > >     - **Synthesizing Dense Crowds**: The reviewer mentions that GCC provides images with thousands of pedestrians. However, we would like to make three points:
> > >         - With regard to the reviewers comment that “thousands of humans, which is the major scene in crowd counting”, we would like to point out that the majority of crowd counting data has fewer than 1000 pedestrians. For example, only 7.5% of images within the JHU++ dataset have 1000+ pedestrians. This is 0% for SHB, 11.0% for SHA, and 21.3% for QNRF.
> > >         -  At inference, we split images into patches for exactly this reason. This is explicitly to transform the images such that they contain crowds with fewer pedestrians. All datasets already have sub-1000 pedestrians for the vast majority of images and we split images further to mitigate this problem.
> > >         -  While our method does not reliably produce images with 1000+ pedestrians, it does produce image pairs where one of the images has 1000+ pedestrians (i.e. the real image). Given this, the model is still exposed to very dense crowds during training.
> > > 3. **Human Labor**: There are two main places where we see human labor used to produce the GCC dataset.
> > >     - **3D assets**: Producing any 3D asset requires some amount of human labor to produce.
> > >     - **Defining Scenes**: The GCC paper ([see here](https://openaccess.thecvf.com/content_CVPR_2019/html/Wang_Learning_From_Synthetic_Data_for_Crowd_Counting_in_the_Wild_CVPR_2019_paper.html)) explains several points where additional labor is required, beyond producing 3D assets:
> > >         - They manually select 100 locations
> > >         - They manually set the parameters of surveillance cameras
> > >         -  Most importantly, they must define regions of interest for 400 scenes for placing pedestrians. We provide a direct quote from the paper: *“Finally, the 400 diverse scenes are built. In these scenes, we elaborately define the Region of Interest (ROI) for placing persons and exclude some invalid regions according to common sense”*.
> > >         - Due to limitations inherent in the GTA V synthesis pipeline, they can only place a maximum of 256 pedestrians. To remedy this, they must merge multiple images. Again, directly quoting the paper *“segment several non-overlapping regions and then place persons in each region”*.
> > >         - Finally, we can see further evidence of this necessary labor clearly demonstrated in the youtube video ([link](https://www.youtube.com/watch?v=Hvl7xWkIueo)) which the authors provided to demo the synthesis process. You can see the authors manually set the camera position. You can also see the authors manually defining the ROI by ***walking a playable character around the scene to define a perimeter. This is an important point, as their process isn’t even as trivial as defining an ROI by simply clicking points. They are manually navigating a playable character around the 3D scene to define the ROI. They do this 400 times.***
> > >
> > > [1] Arteta C, Lempitsky V, Zisserman A.  "Counting in the wild." ECCV 2016
> > > [2] Guerrero-Gómez-Olmedo et al. “Extremely Overlapping Vehicle Counting”. IbPRIA 2015

---

> ### Author Response · Authors · 2023-11-17
> **Table for Noisy Synthetic Crowd Counting statistics**
>
> The table below is produced by using DM-Count [1], which is trained on the NWPU dataset (with full supervision using GT counts), to estimate the count in each noisy synthetic counting image. We report the mean, StD, max, and min count across all synth images. The predicted count can differ significantly from the prompt count for images with higher prompt counts. This suggests that more label noise in larger crowd counts.
> | Prompt Count | Mean   | StD    | Min | Max |
> |--------------|--------|--------|-----|-----|
> | 1            | 0.5   | 1.0   | 0   | 6   |
> | 5            | 1.9   | 1.4   | 0   | 5   |
> | 20           | 18.8  | 11.3  | 2   | 78  |
> | 50           | 71.9  | 59.3  | 10  | 353 |
> | 100          | 150.1 | 118.5 | 19  | 763 |
> | 200          | 245.8 | 93.6  | 37  | 484 |
>
> [1]. Wang, Boyu, et al. "Distribution matching for crowd counting." NeurIPS 2020: 1595-1607.

---

### Meta-Review · Area_Chair_jadP · 2023-12-07

**Metareview:**

The paper initially received mixed ratings of 3,5,6. The major concerns raised by the reviewers were:

1. more explanation about Fig 6 is needed [98cL]
2. what is the advantage of generating data via stable diffusions, compared to GCC [1]? [98cL]
3. [1] also has better counting performance. [98cL]
4. lacking deeper analysis of the method -- does the pre-training method work as expected? [98cL]
5. missing comparison with similar method [2]. [98cL]
6. why do different patch sizes lead to different performance? [98cL]
7. how to set N (crowd count) in synthetic images? [rbvA]
8. only 6 categories of N, then just train a classification model? [rbvA]
9. no N=0 images? [rbvA]
10. what is the computational cost compared to unsupervised counting models? [rbvA]
11. what is the performance of intra-image ranking with ImageNet pre-training [rbvA]
12. does the ranking loss simply train the model to distinguish between real and synthetic images? [rbvA]
13. missing related work on localization methods [QJht]
14. questions about how to handle the unreliable prompt count [QJht]
15. should the original images be cropped into patches during generation? [QJht]
16. no qualitative analysis on the reliability of the generation process [QJht]
17.missing ablation study on fine-tuning the backbone during training [QJht]
18. performance is significantly lower than CrowdCLIP -- what is the result when the method also uses early stop? [QJht]
19. why use image synthesis, when there are existing datasets that can generate ranking image pairs [QJht]

The authors wrote a detailed response. While many points were addressed well, there were a few concerns remaining. First, the performance of SYRAC is inferior to training models on the synthetic dataset GCC [98cL], which is based on GTA5, even without unsupervised domain adaptation. Thus the usefulness of the synthetic data produced by StableDiffusion seems questionable, in particular the lack of localization information of individuals seems to detract from the performance. Authors argued that their approach could be applied to novel categories easily, but only provided some new qualitative results, which was not convincing. Authors also argued that GCC requires human labor to define the scenes and 3d assets, whereas the proposed SYRAC does not. However, SYRAC uses StableDiffusion, which also has a development cost in terms of labor and computation. Where to draw the line is unclear, so the argument is not very convincing. Showing the ease of use to other novel objects might be a better justification.

Second, the performance of SYRAC is inferior to CrowdClip [QJht] on 2/4 datasets. The authors mentioned that they performed model selection using their own synthetic validation set, whereas CrowdClip used the test set for model selection (the oracle evaluation setting). The reviewer asked for results of SYRAC with the oracle evaluation, but these were not provided. During the discussion, the AC noted that many design choices (number of epochs, architecture, optimization method, learning rate, number of patches, etc) are all selected using the test result as reference. Thus these unsupervised methods are generally using the oracle evaluation setting. To give a better reflection of testing without oracle knowledge, it is recommended that works evaluate on the NWPU test set, which is an online hidden evaluation with limited submissions. This may give a better reflection of the performance without oracle knowledge.

The AC agrees with the concerns of the reviewers -- the usefulness of the synthetic data was not convincingly demonstrated. Thus the AC recommended reject.

**Justification For Why Not Higher Score:**

the usefulness of the synthetic data was not convincingly demonstrated -- the performance lags behind other sources of synthetic data or vision-language priors.

**Justification For Why Not Lower Score:**

n/a

---

### Decision · Program_Chairs · 2024-01-16

Reject